# MEGABYTE: Modeling Million-byte Sequences with Multiscale Transformers

**Lili Yu**[*]     **Dániel Simig**[*]     **Colin Flaherty**[*]     **Armen Aghajanyan**

**Luke Zettlemoyer**     **Mike Lewis**

Meta AI

## Abstract

Autoregressive transformers are spectacular models for short sequences but scale poorly to long sequences such as high-resolution images, podcasts, code, or books. We propose MEGABYTE, a multi-scale decoder architecture that enables end-to-end differentiable modeling of sequences of over one million bytes. MEGABYTE segments sequences into patches and uses a *local* submodel within patches and a *global* model between patches. This enables sub-quadratic self-attention, much larger feedforward layers for the same compute, and improved parallelism during decoding—unlocking better performance at reduced cost for both training and generation. Extensive experiments show that MEGABYTE allows byte-level models to perform competitively with subword models on long context language modeling, achieve state-of-the-art density estimation on ImageNet, and model audio from raw files. Together, these results establish the viability of tokenization-free autoregressive sequence modeling at scale.

## 1 Introduction

Sequences of millions of bytes are ubiquitous; for example, music, image, or video files typically consist of multiple megabytes. However, large transformer decoders (LLMs) typically only use several thousand tokens of context (Brown et al., 2020; Zhang et al., 2022a)—both because of the quadratic cost of self-attention but also, more importantly, the cost of large feedforward networks per-position. This severely limits the set of tasks where LLMs can be applied.

We introduce MEGABYTE, a new approach to modeling long byte sequences. First, byte sequences are segmented into fixed-sized patches, loosely analogous to tokens. Our model then consists of three parts: (1) a *patch embedder*, which simply encodes a patch by losslessly concatenating embeddings of each byte, (2) a *global* module, a large autoregressive transformer that inputs and outputs patch representations and (3) a *local* module, a small autoregressive model that predicts bytes within a patch. Crucially, we observe that for many tasks, most byte predictions

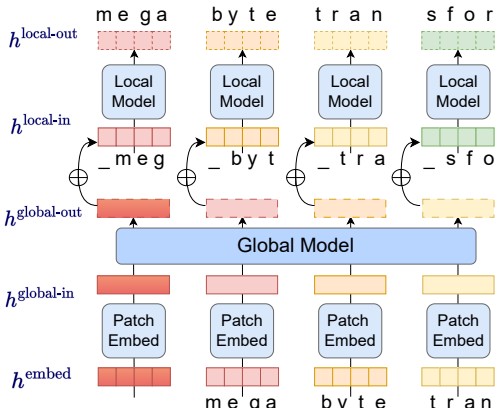

Figure 1: Overview of MEGABYTE with patch size $P = 4$. A small *local* model autoregressively predicts each patch byte-by-byte, using the output of a larger *global* model to condition on previous patches. Global and Local inputs are padded by $P$ and 1 token respectively to avoid leaking information about future tokens.

37th Conference on Neural Information Processing Systems (NeurIPS 2023).

$$h_t^{\text{embed}} = E_{x_t}^{\text{global-embed}} + E_t^{\text{pos}} \qquad\qquad t \in [0..T), E^{\text{global-embed}} \in \mathbb{R}^{V \times D_G},$$

$$E^{\text{pos}} \in \mathbb{R}^{T \times D_G}, h^{\text{embed}} \in \mathbb{R}^{T \times D_G}$$

$$h_k^{\text{global-in}} = \begin{cases} E^{\text{global-pad}}, & \text{if } k = 0, \\ h_{((k-1)\cdot P):(k\cdot P)}^{\text{embed}}, & k \in [1,..,K), \end{cases} \qquad E^{\text{global-pad}} \in \mathbb{R}^{P \times D_G}, K = \frac{T}{P}$$

$$h_{0:K}^{\text{global-out}} = \text{transformer}^{\text{global}}(h_{0:K}^{\text{global-in}}) \qquad\qquad h^{\text{global-out}}, h^{\text{global-in}} \in \mathbb{R}^{K \times P \cdot D_G}$$

$$h_{k,p}^{\text{local-in}} = w^{\text{GL}} h_{k,(p\cdot D_G):((p+1)\cdot D_G)}^{\text{global-out}} + \begin{cases} E^{\text{local-pad}}, & \text{if } p = 0 \\ E_{x_{(k\cdot P + p - 1)}}^{\text{local-embed}}, & p \in [1,..,P) \end{cases} \qquad \begin{aligned} E^{\text{local-pad}} &\in \mathbb{R}^{D_L}, w^{\text{GL}} \in \mathbb{R}^{D_G \times D_L} \\ E^{\text{local-embed}} &\in \mathbb{R}^{V \times D_L} \end{aligned}$$

$$h_{k,0:P}^{\text{local-out}} = \text{transformer}^{\text{local}}(h_{k,0:P}^{\text{local-in}}) \qquad\qquad h_{k,p}^{\text{local-in}} \in \mathbb{R}^{D_L}, h^{\text{local-out}} \in \mathbb{R}^{K \times P \cdot D_L}$$

$$p(x_t | x_{0:t}) = \text{softmax}(E^{\text{local-embed}} h_{k,p}^{\text{local-out}})_{x_t} \qquad\qquad t = k \cdot P + p$$

Figure 2: Summary of MEGABYTE with vocabulary $V$, sequence length $T$, global and local dimensions $D_G$ and $D_L$, and $K$ patches of size $P$. Transformer layers use masked self attention to not observe future timesteps.

are relatively easy (for example, completing a word given the first few characters), meaning that large networks per-byte are unnecessary, and a much smaller model can be used for intra-patch modelling.

MEGABYTE has three main advantages over Transformers for long sequence modeling:

1. **Sub-quadratic self-attention** Most work on long sequence models has focused on mitigating the quadratic cost of self-attention. MEGABYTE decomposes long sequences into two shorter sequences, and optimal patch sizes reduces the self-attention cost to $O(N^{\frac{4}{3}})$, which remains tractable for even long sequences.

2. **Per-patch feedforward layers** In GPT3-size models, more than 98% of FLOPS are used in computing position-wise feedforward layers. MEGABYTE uses large feedforward layers per-patch rather than per-position, enabling much larger and more expressive models for the same cost. With patch size $P$, where a baseline transformer would use the same feedforward layer with $m$ parameters $P$ times, MEGABYTE can use a layer with $mP$ parameters once for the same cost.

3. **Parallelism in Decoding** Transformers must perform all computations serially during generation because the input to each timestep is the output from the previous timestep. By reusing the global representation over multiple time steps during local model decoding, MEGABYTE allows greater parallelism during generation. For example, a MEGABYTE model with 1.5B parameters can generate sequences 40% *faster* than a standard 350M Transformer, whilst also improving perplexity when trained with the same compute.

Together, these improvements allow us to train much larger and better-performing models for the same compute budget, scale to very long sequences, and improve generation speed during deployment.

MEGABYTE also provides a strong contrast to existing autoregressive models that typically use some form of tokenization, where sequences of bytes are mapped to larger discrete tokens (Sennrich et al., 2015; Ramesh et al., 2021; Hsu et al., 2021). Tokenization complicates pre-processing, multi-modal modelling, and transfer to new domains, while hiding useful structure from the model. It also means that most state-of-the-art models are not truly end to end. The most widely used approaches to tokenization require language-specific heuristics (Radford et al., 2019) or lose information (Ramesh et al., 2021). Replacing tokenization with efficient and performant byte models would therefore have many advantages.

We conduct extensive experiments for both MEGABYTE and strong baselines. We use a fixed compute and data budget across all models to focus our comparisons solely on the model architecture rather than training resources, which are known to benefit all models. We find that MEGABYTE allows byte-level models to perform competitively with subword models on long context language modeling, achieve state-of-the-art perplexities for density estimation on ImageNet, and allow audio modelling from raw audio files. Together, these results establish the viability of tokenization-free autoregressive sequence modeling at scale.

## 2  MEGABYTE Transformer

### 2.1  Overview

MEGABYTE is an autoregressive model for efficiently modeling long input sequences. MEGABYTE is comprised of 3 components: (1) a *patch embedder* that inputs a discrete sequence, embeds each element, and chunks it into patches of length $P$ (2) a large *global* Transformer that contextualizes patch representations by performing self-attention over previous patches, and (3) a smaller *local* Transformer that inputs a contextualized patch representation from the global model, and autoregressively predict the *next* patch.

### 2.2  Components

**Patch Embedder** with patch size of $P$ maps a byte sequence $x_{0..T}$ to a sequence of patch embeddings of length $K = \frac{T}{P}$ and dimension $P \cdot D_G$.

First, each byte is embedded with a lookup table $E^{\text{global-embed}} \in \mathbb{R}^{V \times D_G}$ to an embedding of size $D_G$ and positional embeddings are added.

$$h_t^{\text{embed}} = E_{x_t}^{\text{global-embed}} + E_t^{\text{pos}} \qquad\qquad t \in [0..T] \qquad\qquad (1)$$

Then, byte embeddings are reshaped into a sequence of $K$ patch embeddings with dimension $P \cdot D_G$. To allow autoregressive modelling, the patch sequence is padded to start with a trainable patch-sized padding embedding ($E^{\text{global-pad}} \in \mathbb{R}^{P \times D_G}$), and the last patch is removed from the input. This sequence is the input to the global model, and is denoted $h^{\text{global-in}} \in \mathbb{R}^{K \times (P \cdot D_G)}$.

$$h_k^{\text{global-in}} = \begin{cases} E^{\text{global-pad}}, & \text{if } k = 0, \\ h_{((k-1)\cdot P):(k\cdot P)}^{\text{embed}}, & k \in [1,..,K), \end{cases} \qquad\qquad (2)$$

**Global Model** is a decoder-only Transformer with dimension $P \cdot D_G$ that operates on a sequence of $K$ patches. It incorporates a self-attention mechanism and causal masking to capture dependencies between patches. It inputs a sequence of $K$ patch representations $h_{0:K}^{\text{global-in}}$, and outputs an updated representation $h_{0:K}^{\text{global-out}}$ by performing self-attention over previous patches.

$$h_{0:K}^{\text{global-out}} = \text{transformer}^{\text{global}}(h_{0:K}^{\text{global-in}}) \qquad\qquad (3)$$

The output of the final global layer $h_{0:K}^{\text{global}}$ contains $K$ patch representations of dimension $P \cdot D_G$. For each of these, we reshape them into sequences of length $P$ and dimension $D_G$, where position $p$ uses dimensions $p \cdot D_G$ to $(p+1) \cdot D_G$. Each position is then projected to the dimension of the local model with a matrix $w^{\text{GL}} \in \mathbb{R}^{D_G \times D_L}$ where $D_L$ is the local model dimension. We then combine these with byte embeddings of size $D_L$ for the tokens in the *next* patch $E_{x_{(k \cdot P + p - 1)}}^{\text{local-embed}}$. The local byte embeddings is offset by one with a trainable local padding embedding ($E^{\text{local-pad}} \in \mathbb{R}^{D_L}$) to allow autoregressive modelling within a patch. This results in a tensor $h^{\text{local-in}} \in \mathbb{R}^{K \times P \times D_L}$.

$$h_{k,p}^{\text{local-in}} = w^{\text{GL}} h_{k,(p\cdot D_G):((p+1)\cdot D_G)}^{\text{global-out}} + E_{x_{(k \cdot P + p - 1)}}^{\text{local-embed}} \qquad\qquad (4)$$

**Local Model** is a smaller decoder-only Transformer of dimension $D_L$ that operates on a single patch $k$ containing $P$ elements, each of which is the sum of an output from the global model and an embedding of the previous byte in the sequence. $K$ copies of the local models are run on each patch independently (and in parallel during training), computing a representation $h^{\text{local-out}} \in \mathbb{R}^{K \times P \cdot D_L}$.

$$h_{k,0:P}^{\text{local-out}} = \text{transformer}^{\text{local}}(h_{k,0:P}^{\text{local-in}}) \qquad\qquad (5)$$

Finally, we can compute the probability distribution over the vocabulary at each position. The $p$th element of the $k$th patch corresponds to element $t$ of the complete sequence, where $t = k \cdot P + p$:

$$p(x_t | x_{0:t}) = \text{softmax}(E^{\text{local-embed}} h_{k,p}^{\text{local-out}})_{x_t} \qquad\qquad (6)$$

## 2.3 Variations and Extensions

**Convolutional Patch Encoder:** One limitation of patchifying sequences is that it is not translation invariant, and byte sequences may receive a different representation depending on their position in the patch. This may mean, for example, that a model has to relearn the meaning of a word at different offsets. To mitigate this issue, we experimented with augmenting the Patch Embedder with causal convolutional layers, which allow translation-invariant contextual representations of the bytes before they are chunked into patches. We use a stack of convolutional layers, with filter sizes of 3, 5 and 7.

**Cross-patch Attention:** The Local model uses short sequences for efficiency, and relies on the Global model for long-range information. However, we can increase the context of the Local model with little overhead by allowing it to condition on $r$ elements from the previous patch. This approach allows the Global model to focus on a longer-range context. Specifically, when computing self-attention in each layer, we concatenate the keys and values with the last $r$ keys and queries from the previous patch. We use rotary embeddings (Su et al., 2021) to model relative positions between elements in the sequence. This approach is reminiscent of TransformerXL (Dai et al., 2019) but differs by being fully differentiable.

**Strided Inference:** We observed empirically that the per-token loss within each patch increases towards the end of the patch, as the prediction relies more on the weaker Local model. To alleviate this issue, we propose *strided inference*, in which we predict the sequence with two forward passes of the full model, whose inputs are offset by $p/2$ positions from each other. We then combine the first $p/2$ positions in each patch for our predictions to predict the complete sequence. Similarly to sliding window methods (Press et al., 2020), this approach doubles the cost of inference but improves results.

# 3 Efficiency Analysis

## 3.1 Training Efficiency

**Attention** The cost of attention in a transformer architecture for a sequence of length $T$ has $O(T^2)$ complexity. Much work has been explored reducing this; for example, Sparse Transformers (Child et al., 2019) and Routing Transformers (Roy et al., 2020) show strong results with a complexity $O(T^{\frac{3}{2}})$. Many linear attention mechanisms have also been proposed (Katharopoulos et al., 2020; Choromanski et al., 2020), although we are not aware of competitive results on large scale language modeling tasks. As a function of sequence length $T$ and patch size $P$, the Global model has a sequence of length $\frac{P}{T}$ so uses $O(\frac{T^2}{P^2})$ operations, and the Local model uses $\frac{P}{T}$ sequences of length $P$ so uses $O(\frac{TP^2}{P}) = O(PT)$ operations. The overall cost of MEGABYTE is therefore in $O(\frac{T^2}{P^2} + TP)$. $P$ is a hyperparameter that is chosen to create an architecture for sequences of size $T$. By setting $P = T^{\frac{1}{3}}$ the complexity is in $O(T^{\frac{4}{3}})$. Using much shorter patches of $P = T^{\frac{1}{5}}$ would give a complexity of $O(T^{\frac{8}{5}})$. The cost is less than the transformer for all non-trivial values of $P$ such that $1 < P < T$.

**Feedforward Layers** However, attention is not the main cost in large transformers. Instead of increasing the sequence length, transformers are more commonly scaled by increasing the dimension of their latent state $d$, and the feedforward network cost dominates the model's overall cost (Kaplan et al., 2020). For example, in the GPT3 architecture, the quadratic self-attention computation accounts for only 1.4% of FLOPS. Following the approximation of (Kaplan et al., 2020), a forward pass with a large transformer with $m$ non-embedding parameters on a sequence of length $T$ uses roughly $2mT$ FLOPS. MEGABYTE contains two transformers: the Global model uses $m_g$ parameters on a sequence of length $\frac{T}{P}$, and a Local model with $m_l$ parameters that sees $\frac{T}{P}$ sequences of length $P$, giving an estimate of $2T(\frac{m_g}{P} + m_l)$ FLOPS. When $m_g \gg m_l$, the FLOPS used by MEGABYTE is approximately $\frac{2Tm_g}{P}$, allowing a model $P$ times larger than a transformer with equivalent FLOPS. This analysis holds irrespective of any efficient attention mechanisms used in the transformer.

**Combined Analysis** To understand efficiency at different sequence lengths and model sizes, we calculate the total FLOPS used by transformers, Linear Transformers and MEGABYTE. For each operation, we use FLOP estimates from (Kaplan et al., 2020), except for attention in Linear Transformers, which we estimate as $9D$ FLOPS/token[1], where $D$ is the model embedding dimension. Figure 3 shows that for models of size 660M to 173B and sequence lengths of up to 1M tokens,

---

[1]This may underestimate the time taken by Linear Transformer decoders, which use a recurrence mechanism that is harder to parallelize on current hardware.

MEGABYTE with $P = 8$ uses less FLOPS than either transformers or Linear Transformers. Baseline model architectures are based on GPT3, and Megabyte global/local model sizes are 452M/151M, 5.8B/604M, 170B/3.2B respectively.

## 3.2 Generation Efficiency

Generating long sequences with transformers is slow, because the input to each timestep is the output from the previous timestep, meaning each layer must be computed for each token serially. As running a layer on a single token typically does not saturate the amount of parallelism available within a GPU, for analysis, we model each layer as a constant cost independently of size. Consider a MEGABYTE model with $L_{\text{global}}$ layers in the Global model and $L_{\text{local}}$ layers in the Local model and patch size $P$, compared with a Transformer architecture with $L_{\text{local}} + L_{\text{global}}$ layers. Generating each patch with MEGABYTE requires a sequence of $O(L_{\text{global}} + P \cdot L_{\text{local}})$ serial operations, whereas the Transformer requires $O(P \cdot L_{\text{global}} + P \cdot L_{\text{local}})$ serial operations. When $L_{\text{global}} \gg L_{\text{local}}$ (i.e. the Global model has many more layers than the Local model), MEGABYTE can reduce inference costs by a factor close to $P$.

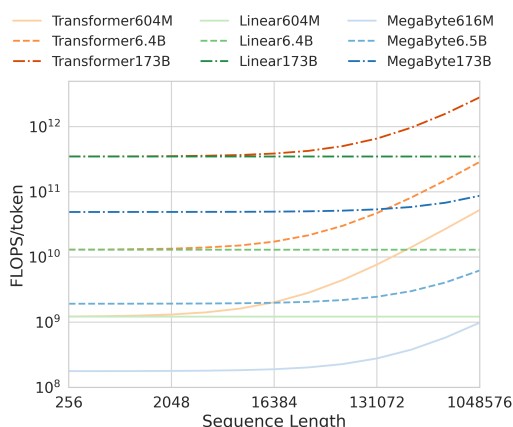

Figure 3: Computational cost (FLOPS/token) for different model architectures at different scales. MEGABYTE architectures (here with $P = 8$) use less FLOPS than equivalently sized Transformers and Linear Transformers (Katharopoulos et al., 2020) across a wide range of model sizes and sequence lengths, allowing larger models to be used for the same computational cost.

## 4 Experimental setup

**Controlling for Compute and Data** Models show consistent improvements when increasing both data and compute Kaplan et al. (2020); Hoffmann et al. (2022), meaning that one model can outperform another because of an increased training budget instead of an improved architecture. However, in practice, both compute and data are typically limited. We conduct experiments using a fixed compute and data budget across all models to focus comparisons solely on the model architecture rather than training resources. To achieve this, we adjust model hyperparameters (mainly, number of layers) within each architecture so that the forward pass time taken per byte is matched, and then train all models for the same number of bytes.

**Comparison Systems** We compare MEGABYTE with both a standard decoder-only Transformer and PerceiverAR (Hawthorne et al., 2022). PerceiverAR extends the original transformer with a single cross-attention layer over a much longer context sequence, and is the best performing general purpose autoregressive model we are aware of and achieves state-of-the-art results across several modalities. We implemented both models in the same codebase, and all models share a similar data loader, preprocessing step, and trainer to avoid any artifacts in our compute-controlled experiments.

**Training Procedure** All models were trained using the Metaseq[2] code base Zhang et al. (2022b). The training used the PyTorch framework Paszke et al. (2019), with fairscale to improve memory efficiency through fully sharded model and optimizer states Baines et al. (2021). Mixed precision training was used to improve training efficiency at scale Micikevicius et al. (2017). More training details and various model parameters can be found in Section A.1 in the Appendix. To validate our implementation of PerceiverAR, we reproduced their experiments on downsized ImageNet at 64 pixels. By carefully matching hyperparameters, we achieved a bits per byte (bpb) score of 3.53, compared to the reported 3.54 in the original paper.

**Inference Methods** Several techniques have been proposed for trading off speed for performance during inference with language models, including sliding windows Press et al. (2020) and our strided inference. We only use these methods when comparing with prior published work (Tables 2 and 3).

---

[2]https://github.com/facebookresearch/metaseq

| Dataset | Total Bytes | bytes/doc | Transformer | PerceiverAR | MEGABYTE |
|---------|-------------|-----------|-------------|-------------|----------|
| PG-19   | 10.1GB      | 411,404   | 1.057       | 1.104       | **1.000** |
| Stories | 21.3GB      | 35,265    | 1.064       | 1.070       | **0.978** |
| Books   | 79.7GB      | 509,526   | 1.097       | 1.104       | **1.007** |
| arXiv   | 91.5GB      | 58,518    | 0.816       | 0.791       | **0.678** |
| Code    | 353.7GB     | 7,461     | 0.575       | 0.546       | **0.411** |

Table 1: Text dataset sizes and mean document lengths. We also report bpb of various models (Transformer, PerceiverAR, and MEGABYTE) trained with the same compute.

| | Tokenizer | Vocab | Context Length | Validation | Test |
|---|-----------|-------|----------------|------------|------|
| TransformerXL Rae et al. (2019a) | SentPiece | 32k | 512+1024 | 45.5 | 36.3 |
| CompressiveTransformer Rae et al. (2019a) | SentPiece | 32k | 512+512+2x512 | 43.4 | 33.6 |
| PerceiverAR Hawthorne et al. (2022) | SentPiece | 32k | 2048 | 45.9 | 28.9 |
| BlockRecurrent Hutchins et al. (2022) | SentPiece | 32k | 1024+recurrence | - | **26.5** |
| Transformer byte-level (ours) | Bytes | 256 | 2048 | 81.6 | 69.4 |
| PerceiverAR byte-level (ours) | Bytes | 256 | 8192 | 119.1 | 88.8 |
| MEGABYTE | Bytes | 256 | 8192 | **42.8** | 36.4 |

Table 2: Larger scale experiments on PG19, converting bits-per-byte to word-level perplexities for comparison with prior work. Results below the line are compute-matched. MEGABYTE outperforms other byte models by a wide margin, and gives results competitive with state-of-the-art models trained on subwords.

## 5   Language Modeling

We evaluated the performance of MEGABYTE on language modeling on a set of 5 diverse datasets emphasizing long-range dependencies: Project Gutenberg (PG-19), Books, Stories, arXiv, and Code.

**Datasets** We experiment on a range of long form text datasets. The PG-19 dataset Rae et al. (2019b) consists of English-language books written before 1919 and is extracted from the Project Gutenberg online library. The Stories dataset Trinh & Le (2018) is a subset of CommonCrawl data meant to emulate Winograd schemas. Books Gao et al. (2020) is another collection of English-language books. The arXiv dataset contains technical publications written in LATEX from the arXiv online archive. Finally, the Code dataset is a large publicly available dataset of open source code, under Apache, BSD or MIT licenses. More details on dataset sizes and document lengths are shared in Table 1.

**Controlled Experiments** Table 1 lists bpb on each dataset. Each model is trained for 80 billion bytes, and models are scaled to use the same compute budget. We carefully tune hyperparameters for all architectures to best utilize the available compute budget. MEGABYTE consistently outperforms both transformers and PerceiverAR across all datasets. We use the same sets of parameters on all dataset. In all experiments presented in Table 1, transformer has size of 320M with context length of 1024, PerceiverAR has size of 248M with context size of 8192 and latent size of 1024, and MEGABYTE global/local model sizes are 758M/262M with context length of 8192 and patch size of 8.

**Scaling Experiment** We scale up our training data on PG-19 (Table 2), and compare MEGABYTE with byte baselines, as well as converting all results to word-level perplexities to benchmark with state-of-art token based models. We train a byte-level Transformer, PerceiverAR and MEGABYTE models for 400B bytes and the same compute budget using same model parameters as in the controlled experiments. We find that MEGABYTE outperforms other byte-level models by a wide margin at this scale.[3] We also compare with the best previously reported numbers for sub-word models. These results may be confounded by differing amounts of compute and tuning used, but show that MEGABYTE gives results competitive with state-of-the-art models trained on subwords. These results suggest that MEGABYTE may allow future large language models to be tokenization-free.

## 6   Image Modeling

**Sequence Modeling on ImageNet** We test MEGABYTE on variants of the autoregressive image generation task on ImageNet (Oord et al., 2016), to measure its ability to efficiently use long context. We test on three different resolutions of images, ranging from 64×64 to 640×640 pixels – the latter

---

[3]The only prior byte-level experiments we are aware of are at a smaller scale in Hutchins et al. (2022), who report results equivalent to test perplexities of 46.5 with a version of the BlockRecurrent transformer, and 49.5 with Memorizing Transformers Wu et al. (2022), compared to 36.4 with our model.

requiring the effective modeling of sequences with over 1.2M tokens. This generation task becomes increasingly challenging as the image's resolution grows: doing well on this task requires the modeling of local patterns (textures, lines, etc.) and long-range context that provides information about the high level structure of the image. Inspired by recent works in Vision Transformers (Dosovitskiy et al., 2020), we model image data patch by patch (more details can be found in Appendix D.1).

**Comparison with State of the Art**   We train a large MEGABYTE model on ImageNet 64x64 with Global and Local models sized 2.7B and 350M parameters, respectively, for 1.4T tokens. We estimate that training this model consumed less than half the GPU hours we would have needed to reproduce the best PerceiverAR model described by (Hawthorne et al., 2022). As shown in Table 2, MEGABYTE matches the state-of-the-art performance of PerceiverAR whilst using only half the compute.

| ImageNet64 | bpb |
|---|---|
| Routing Transformer (Roy et al., 2020) | 3.43 |
| Combiner (Ren et al., 2021) | 3.42 |
| Perceiver AR (Hawthorne et al., 2022) | **3.40** |
| MEGABYTE | **3.40** |

Table 3: Bits per byte (bpb) on ImageNet 64×64. MEGABYTE matches the current state-of-the-art while only using half the amount of GPU hours to train.

| | Context | Image64 | Image256 | Image640 |
|---|---|---|---|---|
| Total len | | 12288 | 196608 | 1228800 |
| Transformer | 1024 | 3.62 | 3.801 | 2.847 |
| Perceiver AR | 12000 | 3.55 | 3.373 | 2.345 |
| MEGABYTE | Full | **3.52** | **3.158** | **2.282** |

Table 4: Bits per byte (bpb) on ImageNet with different resolutions. All models use the same compute and data. MEGABYTE scales well to sequences of over 1M tokens.

**Scaling to higher resolutions**   We compare three transformer variants (vanilla, PerceiverAR, MEGABYTE) to test scalability to long sequences on increasingly large image resolutions. We use our own implementations of these in the same framework and budget the same amount of GPU hours and data to train each of these model variants.

MEGABYTE is able to handle all sequence lengths with a single forward pass of up to 1.2M tokens. We found neither the standard Transformer nor PerceiverAR could model such long sequences at a reasonable model size, so instead we split images into segments of size 1024 and 12000 respectively. For Megabyte, we set patch size as 12 for Image64 and patch size as 192 for Image256 and Image640 datasets. Model sizes are adjusted to match overall training speeds across models and we do not use any form of sliding window evaluation in this experiment. As seen in Table 4, MEGABYTE outperforms baselines across all resolutions in this compute-controlled setting. The precise settings used for each of the baseline models such as context length and number of latents are summarized in Table 12. Results show that MEGABYTE outperforms the other systems at all resolutions, demonstrating an effective model of sequences of over 1M bytes.

# 7   Audio Modeling

Audio has aspects of both the sequential structure of text and the continuous nature of images, so is an interesting application for MEGABYTE.

Raw audio is typically stored as a sequence of 16-bit integer values (one per timestep); a softmax layer would need to output 65,536 probabilities per timestep to model all possible values. To address this issue, various techniques have been developed to reduce the memory and computational requirements of the softmax layer. For instance, van den Oord et al. (2016) apply $\mu$-law companding transformation and quantizes the input into 256 possible values. Alternatively, van den Oord et al. (2017) model the samples using the discretized mixture of logistics distribution introduced by Salimans et al. (2017). Finally, Kalchbrenner et al. (2018) use a dual softmax technique to produce 8 coarse and 8 fine bits. In our approach, we simplify the audio modeling process by directly reading the bytes (256 possible values) from the audio file and conducting an autoregressive language model on top of that. This greatly streamlines the modeling process, making it easier and more efficient.

Our audio modeling approach focuses on 16 kHz, 16-bit audio, which equates to 32k bytes per one-second clip. We use an extensive audio dataset consisting of 2 terabytes (roughly 18,000 hours) of audio. We use a sequence length of 524,288, a patch size of 32, and a batch size of 32 to facilitate model training. By utilizing these settings, we can effectively train our model on large volumes of audio data, helping to improve its accuracy and efficacy. Our model obtains bpb of 3.477, much

|  | Global Size | (Local) Size | bpb | Generation Time (s) |
|---|---|---|---|---|
| Transformer | - | 350M | 1.064 | 132 |
| MEGABYTE | 1.3B | 218M | 0.991 | 93 |

Table 5: Comparison of bits per byte (bpb) and generation speed of 8192 bytes of transformer model (with context length 1024) and MEGABYTE with context length 8192 and patch size 8.

lower than the results with perceiverAR (3.543) and vanilla transformer model (3.567). More ablation results are presented in Table 6.

## 8 Analysis

We study different behaviors of MEGABYTE. All experiments in the same group use the same compute.

**Generation speed** We also compare the text generation speed between MEGABYTE and a transformer. We compare a 350M parameter baseline transformer and a MEGABYTE model with a 1.3B parameter Global model and a 218M parameter local model, trained on PG19 with equal compute. As shown in Table 5, the MEGABYTE model achieves much lower perplexity as expected. However, MEGABYTE also generates a sequence of 8192 tokens 40% *faster* than transformer, despite having over 4 times the parameters. This speed up is due to the bulk of the parameters being in the Global model, which only needs to be computed once for every 8 tokens, whereas all the parameters in the baseline model are used on every token.

**Model Components** In Table 6, we analyze the significance of different components in the MEGABYTE architecture by studying arXiv, Librilight-L and ImageNet256 datasets. Removing Local (*w/o local model*) or global (*w/o global model*) model, we observe a substantial increase in bpb on all datasets, showing that both parts are crucial. The performance of the model without the cross-patch local model (*w/o cross-patch local model*) is competitive, indicating that the architecture is robust to this modification. We observe slight improvement on the Librilight-L and ImageNet256 datasets by augmenting the MEGABYTE model with a CNN encoder (*w/ CNN encoder*). This suggests that the MEGABYTE architecture can benefit from integrating alternative encoding mechanisms.

**Effective Use of Context** Long-context models often struggle to benefit from the full context (Sun et al., 2021). Figure 7 shows that later tokens within each context window have a higher likelihood, indicating that MEGABYTE can effectively use at least 8k bytes of context on the PG19 dataset.

|  | Arxiv | Audio | Image256 |
|---|---|---|---|
| MEGABYTE | 0.6871 | **3.477** | **3.158** |
| *w/o local model* | 1.263 | 5.955 | 4.768 |
| *w/o global model* | 1.373 | 3.659 | 3.181 |
| *w/o cross-patch attention* | **0.6781** | 3.481 | 3.259 |
| *w/ CNN encoder* | 0.6871 | **3.475** | **3.155** |

Table 6: Ablation of MEGABYTE model components. Models with the same dataset are trained using the same compute. The hyperparameters are listed in Table 12.

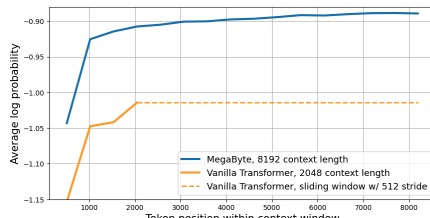

Table 7: Average log probability assigned to different positions within the context length by MEGABYTE and by a vanilla transformer model on PG19 test set.

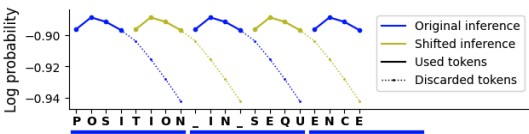

Table 8: An illustration of strided inference with patch size 8. Blue and yellow represents two inferences that are shifted by half patch size. Solid line indicates final probablity being taking during strided inference.

| Method | Inference Cost | bpb |
|---|---|---|
| Basic Inference | 1X | 0.9079 |
| *w/ Sliding Window* | 2X | 0.8918 |
| *w/ Strided Inference* | 2X | 0.8926 |
| *w/ Sliding & Strided* | 4X | **0.8751** |

Table 9: Performance of various inference techniques on the PG19 test set using our best MEGABYTE model.

**Strided Inference** We find that within a single patch, on average, the MEGABYTE performs worse on later tokens within a patch (see Figure 8). Section 2.3 proposes *strided inference* as a solution, where two forward passes are performed offset by $\frac{P}{2}$ tokens, and results from the first half of each patch are combined. Table 9 shows performance improvements from strided inference, which are additive with the standard sliding window.

**Patch Size.** We experimented with various patch sizes on Image256 dataset and found a wide range of values where MEGABYTE performs similarly. We found similar robustness to patch size choices across all modalities, although the optimal patch size itself can be different across modalities.

**Local to Global model Size Ratio.** We experimented with different Local/Global model size ratios on PG19 dataset. By grouping bytes into patches, MEGABYTE effectively uses $P$ times less tokens for the Global model as on the Local model—enabling us to increase the size of the Global model with reduced cost. We find that a given compute budget is spent optimally when the Global model is larger than the Local model, consistently across all modalities and various patch sizes.

| Patch | Global Size | Local Size | bpb |
|---|---|---|---|
| 48 | 125M | 114M (L=11) | 3.178 |
| 192 | 125M | 125M (L=12) | 3.158 |
| 768 | 125M | 83M (L=8) | 3.186 |

Table 10: Effects of patch size on performance on the Image256 dataset. All versions use the same amount of GPU hours and data.

| Global Size | Local Size | bpb |
|---|---|---|
| 350M (D=1024,L=24) | 290M (D=1024,L=20) | 1.014 |
| 760M (D=1536,L=24) | 262M (D=1024,L=18) | 1.002 |
| 1.3B (D=2048,L=24) | 218M (D=1024,L=15) | 0.991 |

Table 11: Effects of Local / Global model size on the PG19 dataset. Increasing the capacity of global model improves performance. Models are compute and data matched.

# 9 Related Work

Prior research has explored the possibility of improving the efficiency of Transformers on long sequences, primarily motivated by mitigating the quadratic cost of self-attention.

**Efficient Encoder Models** Several related techniques to ours have been developed for transformer encoder architectures but cannot be straightforwardly applied to decoders. In particular, patchifying operations have previously been used in image *encoder* models such as ViT (Dosovitskiy et al., 2020), and down- and up-sampling operations have been used for text encoders (Clark et al., 2022), but such methods cannot be naively applied to decoder-only models without leaking information to future bytes in the same patch. MEGABYTE generalizes these approaches to an efficient decoder model by using a intra-patch transformer to predict each sequence element's likelihood, and offsetting the inputs to the two models to avoid leaking information. Jaegle et al. (2021) use self-attention on a shorter latent sequence also resembles patchification, but this technique cannot easily be applied to decoder architectures without leaking information to future timesteps.

**Efficient Decoder models** Improving the efficiency of decoder models is harder because of the need to make one prediction per timestep, and not leak information to future timesteps. The most popular approaches can be categorized as (1) chunking sequences into smaller blocks, and propagating information from previous blocks with either recurrence (Dai et al., 2019; Hutchins et al., 2022) or cross-attention (Hawthorne et al., 2022), (2) linear alternatives to attention, which typically involve forms of token-level recurrence (Katharopoulos et al., 2020) or state space models (Gu et al., 2021; Smith et al., 2022; Ma et al., 2022), or (3) sparse approximations of attention (Kitaev et al., 2020; Beltagy et al., 2020; Child et al., 2019; Wu et al., 2022). However, the performance of dense attention means it is typically still chosen for large scale decoders (Touvron et al., 2023; Chowdhery et al., 2022). MEGABYTE takes the alternative approach of decomposing the complete sequence into two shorter sequences, giving sub-quadratic attention. We also note that feedforward networks are the dominant cost in large decoders, not self-attention. Our approach to compressing sequences allows much larger models than would be possible when using large feedforward networks at every timestep.

**Tokenization** The most common approach to shortening sequence lengths in Transformer decoders is to pre-process the input with a form of tokenization, in which multiple bytes are mapped to a single discrete token from a fixed vocabulary. For text, this can be done losslessly using methods such as BPE (Sennrich et al., 2015) and SentencePiece (Kudo & Richardson, 2018), but these approaches can require language-specific heuristics (Radford et al., 2019), limit out-of-domain performance (Sharami

et al., 2023), and can affect prompting and truncated sampling in unpredictable ways.[4] The amount of high-frequency information in images and audio means that tokenization cannot be performed losslessly, and instead clustering (Hsu et al., 2021) or discrete auto-encoders (Ramesh et al., 2021) are used to compress the inputs, which lose information and likely limit generative model performance. Our patches are analogous to traditional lossless tokens, and the Local model performs the role of mapping a hidden state to a distribution over possible patches.

## 10 Conclusion

We introduced MEGABYTE, a scaleable architecture for modeling long sequences. MEGABYTE outperforms existing byte-level models across a range of tasks and modalities, allowing large models of sequences of over 1 million tokens. It also gives competitive language modeling results with subword models, which may allow byte-level models to replace tokenization. However, the scale of experiments here is far below those of state-of-the-art language models (Brown et al., 2020), and future work should explore scaling MEGABYTE to much larger models and datasets.

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

# A Supplementary Material

## A.1 Training Details

To ensure stable training, we applied gradient clipping with a maximum norm of 1.0 and used the Adam optimizer with $\beta_1 = 0.9$, $\beta_2 = 0.98$ Kingma & Ba (2015). We used the built-in polynomial decay learning rate scheduler in MetaSeq with 500 warmup updates and the end learning rate set to 0. All models are trained with pre-norm and using ReLU activation. We apply a dropout of 0.1 throughout, but we do not apply any dropout to embeddings. We also use weight decay of 0.1. To initialize the weights, we use a variant based on Megatron-LM codebase, which involves using a normal distribution with a mean of zero and a standard deviation of 0.006. We truncate this normal distribution within two standard deviations and observed substantial gain in both training stability and performance.

## A.2 Motivation

**Why is the local model needed?** Many of the efficiency advantages of the MEGABYTE design could be realized with the Global model alone, which would resemble a decoder version of ViT (Dosovitskiy et al., 2020). However, the joint distribution over the patch $p(x_{t+1}, .., x_{t+P}|x_{0..t})$ has an output space of size $256^P$ so direct modeling is only tractable for very small patches. We could instead factor the joint distribution into conditionally independent distributions $p(x_{t+1}|x_{0..t})..p(x_{t+P}|x_{0..t})$, but this would greatly limit the model's expressive power. For example, it would be unable to express a patch distribution such as 50% *cat* and 50% *dog*, and would instead have to assign probability mass to strings such as *cag* and *dot*. Instead, our autoregressive Local model conditions on previous characters within the patch, allowing it to only assign probability to the desired strings.

**Increasing Parameters for Fixed Compute** Transformer models have shown consistent improvements with parameter counts (Kaplan et al., 2020). However, the size of models is limited by their increasing computational cost. MEGABYTE allows larger models for the same cost, both by making self attention sub-quadratic, and by using large feedforward layers across patches rather than individual tokens.

**Re-use of Established Components** MEGABYTE consists of two transformer models interleaved with shifting, reshaping and a linear projection. This re-use increases the likelihood that the architecture will inherit the desirable scaling properties of transformers.

## A.3 Model Details

As discussed in Section 4, we conduct experiments using a fixed compute and data budget across all models to focus our comparisons solely on the model architecture rather than training resources. To achieve this, we adjust model hyperparameters within each architecture so that the time taken for a single update is matched and then train all models for the same number of updates. We list all of model details in Table 12 and Table 13.

|  | Model | #L | $d_{\text{model}}$ | #H | $d_{\text{head}}$ |
|---|---|---|---|---|---|
| S1 | 125M | 12 | 768 | 12 | 64 |
| S2 | 350M | 24 | 1024 | 16 | 64 |
| S3 | 760M | 24 | 1536 | 16 | 96 |
| S4 | 1.3B | 24 | 2048 | 32 | 64 |
| S5 | 2.7B | 32 | 2560 | 32 | 80 |
| S6 | 6.7B | 32 | 4096 | 32 | 128 |

Table 12: **Common Model architecture details by size.** For each model size, we show the number of layers (#L), the embedding size ($d_{\text{model}}$), the number of attention heads (#H), the dimension of each attention head ($d_{\text{head}}$).

| Model | (Global) Size | Local Size | BS | LR | Context Length (in bytes) |
|---|---|---|---|---|---|
| arXiv | | | | | |
| Transformer | 320M (D=1024, L=22) | N/A | 72 | 2.00E-04 | 1,024 |
| Perceiver AR | 248M (D=1024, L=17) | N/A | 72 | 2.00E-04 | 8,192 (1024 latents) |
| MEGABYTE | 758M (D=2048, L=14) | 262M (D=1024, L=18) | 48 | 2.00E-04 | 8,192 (patch size 8) |
| *w/o Local model* | 2.3B (D=2560, L=20) | N/A | 48 | 1.50E-04 | 8,192 (patch size 4) |
| *w/o global model* | N/A | 350M (D=1024, L=24) | 192 | 2.00E-04 | 8,192 (patch size 8) |
| *w/o cross-patch Local model* | 921M (D=2048, L=17) | 350M (D=1024, L=24) | 48 | 2.00E-04 | 8,192 (patch size 8) |
| *w/ CNN encoder* | 704M (D=2048, L=13) | 262M (D=1024, L=18) | 48 | 2.00E-04 | 8,192 (patch size 8) |
| Image task 64 (Table 2) | | | | | |
| MEGABYTE | 2.7B (D=2560, L=32) | 350M (D=1024, L=24) | 2 | 2.00E-04 | 12,288 (patch size 12) |
| Image task 64 (Table 4) | | | | | |
| Transformer | 760M (D=1536, L=24) | N/A | 512 | 3.00E-04 | 2,048 |
| Perceiver AR | 227M (D=1024, L=16) | N/A | 512 | 3.00E-04 | 12,288 (1024 latents) |
| MEGABYTE | 1.3B (D=2048, L=24) | 1.3B (D=2048, L=24) | 256 | 3.00E-04 | 12,288 (patch size 12) |
| Image task 256 | | | | | |
| Transformer | 62M (D=768, L=6) | N/A | 1536 | 2.00E-04 | 1,024 |
| Perceiver AR | 62M (D=768, L=6) | N/A | 256 | 2.00E-04 | 8,192 (768 latents) |
| MEGABYTE | 125M (D=768, L=12) | 125M (D=768, L=12) | 16 | 2.00E-04 | 196,608 (patch size 192) |
| *w/o local model* | 2.7B (D=4096, L=32) | N/A | 16 | 2.00E-04 | 196,608 (patch size 48) |
| *w/o global model* | 125M (D=768, L=12) | 125M (D=768, L=12) | 16 | 2.00E-04 | 196,608 (patch size 192) |
| *w/o cross-patch Local model* | 250M | 156M (D=768, L=15) | 16 | 2.00E-04 | 196,608 (patch size 192) |
| *w/ CNN encoder* | 125M (D=768, L=12) | 125M (D=768, L=12) | 16 | 2.00E-04 | 196,608 (patch size 192) |
| Image task 640 | | | | | |
| Transformer | 83M (D=768, L=8) | N/A | 4800 | 3.00E-04 | 1,024 |
| Perceiver AR | 62M (D=768, L=6) | N/A | 2048 | 3.00E-04 | 4,096 (1024 latents) |
| MEGABYTE | 125M (D=768, L=12) | 83M (D=768, L=8) | 32 | 3.00E-04 | 1,228,800 (192 patch size) |
| audio | | | | | |
| Transformer | 135M (D=768, L=13) | N/A | 2048 | 2.00E-04 | 1024 |
| Perceiver AR | 62M (D=768, L=6) | N/A | 384 | 2.00E-04 | 8,192 (1024 latents) |
| MEGABYTE | 350M (D=1024, L=24) | 125M (D=768, L=12) | 256 | 2.00E-04 | 524,288 (32 patch size) |
| *w/o local model* | 2.7B (D=4096, L=32) | 125M (D=768, L=12) | 256 | 2.00E-04 | 524,288 (32 patch size) |
| *w/o global model* | 350M (D=1024, L=24) | 125M (D=768, L=12) | 256 | 2.00E-04 | 524,288 (32 patch size) |
| *w/o cross-patch Local model* | 350M (D=1024, L=24) | 146M (D=768, L=14) | 256 | 2.00E-04 | 524,288 (32 patch size) |
| *w/ CNN encoder* | 350M (D=1024, L=24) | 125M (D=768, L=12) | 256 | 2.00E-04 | 524,288 (32 patch size) |

Table 13: **Model architecture details.** We report the model size, the embedding size (D), number of layaers(L), total batch size (BS), learning rate(LR), and context length. When we vary the number of model layers from the standard amount for the given size (Table 12), we note this accordingly. For PerceiverAR models, we note the number of latents used, and for MEGABYTE models we note the patch sizes.

# B  Pseudocode

Listing 1: Pseudocode of Megabyte model

```python
class MegaByteDecoder:
    def __init__(
        self,
        global_args,
        local_args,
        patch_size,
    ):
        self.pad = 0
        self.patch_size = patch_size
        self.globalmodel = TransformerDecoder(global_args)
        self.localmodel = TransformerDecoder(local_args)

    def forward(
        self,
        bytes,
    ):
        bytes_global, bytes_local = self.prepare_input(bytes)
```

```python
        global_bytes_embedded = self.globalmodel.embed(bytes_global)
        global_in = rearrange(
            global_bytes_embedded,
            "b (t p) e -> b t (p e)",
            p=self.patch_size,
        )
        global_output = self.globalmodel(global_in)

        global_output_reshaped = rearrange(
            global_output,
            "b t (p e) -> (b t) p e",
            p=self.patch_size,
        )
        local_bytes_embedded = self.localmodel.embed(bytes_local)
        local_in = local_bytes_embedded + global_output_reshaped
        local_output = self.localmodel(local_in)

        batch_size = bytes_global.shape[0]
        x = rearrange(local_output, "(b t) l v  -> b (t l) v", b=
            batch_size)
        return x

    def prepare_input(self, bytes):
        padding_global = bytes.new(bytes.shape[0], self.patch_size).
            fill_(self.pad)
        bytes_global = torch.cat((padding_global, bytes[:, : -self.
            patch_size]), -1)

        bytes_input = rearrange(bytes, "b (t p) -> (b t) p", p=self.
            patch_size)
        padding_local = bytes_input.new(bytes_input.shape[0], 1).fill_
            (self.pad)
        bytes_local = torch.cat((padding_local, bytes_input[:, :-1]),
            -1)

        return bytes_global, bytes_local
```

## C  PerceiverAR Implementation

To reproduce PerceiverAR in a compute-controlled setting we extended the standard transformer implementation in metaseq with an additonal cross attention layer to compute the latents and match the architecture of PerceiverAR. We trained the model by sampling random spans from each text, matching the procedure used in the PerceiverAR codebase. To be consistent with the original work, we use sliding window evaluation with a stride of $num\_latents/2$ unless otherwise noted. In several cases we used the standard metaseq implementation as opposed to specific techniques reported in the original paper: 1) we used standard attention dropout instead of cross-attention dropout 2) We did not implement chunked attention. We verified our implementation by reproducing the "Standard Ordering" experiments in Table 5 of the Perceiver AR paper. After carefully matching context size, number of latents, the amount of data and training steps used and learning rate, we achieved 3.53 bpb vs 3.54 reported in the original paper.

## D  More results

### D.1  Patch scan Implementation

Images have a natural structure, containing a grid of $n \times n$ pixels each composed of 3 bytes (corresponding to color channels). We explore two ways of converting images to sequences for modeling (see Figure 4). Firstly, *raster scan* where the pixels are linearized into 3 bytes and concatenated row-by-row. Secondly, *patch scan* where we create patches of shape $p \times p \times 3$ bytes

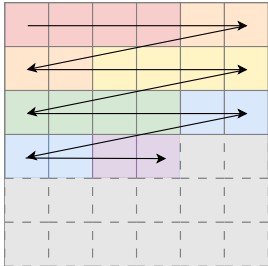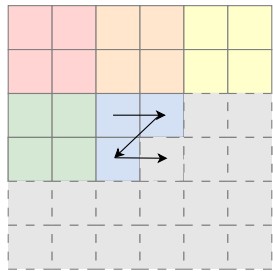

Figure 4: Two ways to model 2D data sequentially. Left, raster scan, by taking bytes row by row and left to right; right, patch scan, where we first split an image into patches, and do raster scan across patches and within a patch. (T=36, K=9, P=4).

where $p = \sqrt{\frac{P}{3}}$, and then use a raster scan both within and between patches. Unless otherwise specified, MEGABYTE models use *patch scan* for image data.

### D.2 Patch scan vs Raster scan

The patch scan method is inspired by recent works in Vision Transformers (Dosovitskiy et al., 2020), and it is more effective than raster scan for modeling image sequencing. We found it improves both MEGABYTE and Perceiver AR.

|  | (Global) Size | Local Size | context | bpb |
|---|---|---|---|---|
| MEGABYTE (patch scan) | 62M (D=768, L=6) | N/A | 8,192 (768 latents) | 3.158 |
| MEGABYTE (raster scan) | 62M (D=768, L=6) | N/A | 8,192 (768 latents) | 3.428 |
| Perceiver AR (patch scan) | 125M (D=768, L=12) | 125M (D=768, L=12) | 196,608 (patch size 192) | 3.373 |
| Perceiver AR (raster scan) | 125M (D=768, L=12) | 125M (D=768, L=12) | 196,608 (patch size 192) | 3.552 |

Table 14: ImageNet256 performance with patch scan vs raster scan for MEGABYTE and Perceiver AR.

### D.3 Longer sequence modeling

For our pg19 scaling experiment, we also use longer context length for MEGABYTE. The results are shown in Table 15. With longer sequence, we didn't observer further improvement, consistent with findings in Hawthorne et al. (2022). We think we will benefit more from longer sequence when we futher scale up the model size and data.

|  | context | bpb |
|---|---|---|
| MEGABYTE | 8,192 (patch size 8) | 0.8751 |
| MEGABYTE | 16,384 (patch size 8) | 0.8787 |

Table 15: Longer sequence for PG19 dataset. For both experiments, we set global model as 1.3b, local model as 350m, and MEGABYTE patch size as 8.

