# OpenReview forum: "MEGABYTE: Predicting Million-byte Sequences with Multiscale Transformers"
_NeurIPS.cc/2023/Conference — NeurIPS 2023 poster_

### Official Review · Reviewer_W4cB · 2023-06-10

**Soundness:** 2 fair
**Presentation:** 4 excellent
**Contribution:** 4 excellent
**Rating:** 7
**Confidence:** 4

**Summary:**

The authors propose MegaByte, a decoder style model consisting of 3 parts: patching of bytes, processing of patches (with autoregressive model), and converting patches back to bytes (with autoregressive model). Their model allows for sub-quadratic attention, more expressive models for the same flop count, faster decoding, and no need for tokenization. They show that it outperforms Transformer and Perceiver AR on byte-level modeling of language, images, and audio. There is also some indication that future language models may be able to successfully avoid tokenization entirely by using this model type.

**Strengths:**

**Originality**: The proposed model architecture is novel. Comparison to Perceiver AR seems reasonable and shows understanding of related work in the area.

**Quality**: Experiments do a good job of demonstrating the quality of the model.

**Clarity**: For the most part the paper is well-written and easy to follow.

**Significance**: This paper brings forward more evidence that tokenization could be avoided entirely in language modeling. It explores a well-motivated and novel architecture. It opens up some interesting possibilities for future work.

**Weaknesses:**

Main weakness: While the paper does a laudable job of covering a lot of ground, this comes at a cost.
1. First, the paper is unclear about when the three "Extensions" proposed in Section 2.3 are used in the results tables. It would be great to have something like a table in the appendix that just tells for each MegaByte result whether each extension was used. That would help readers get a sense (beyond Figure 4) of how much heavy lifting the architecture is doing vs the extensions.
2. The paper focuses on high-level ideas (as it should) but it leaves some low-level details unclear. Here are a couple examples:
    * Does the local model use position embeddings (when not using cross-patch attention)?
    * What audio modeling dataset was used?
    * When the GPT-3 architecture is brought up, is that including use of locally banded sparse attention patterns?
3. In general, the results in the paper are not reproducible.

With the above being said, I do want to point out that the authors have clearly made efforts along these lines as the appendix has substantial detail.

A quick fix to all of my above concerns would be to (1) release code associated with the paper, and (2) to add more detail regarding when "Extensions" were used in the paper.

**Questions:**

Please see weaknesses.

**Limitations:**

There is currently no substantive discussion of limitations, and it could be a helpful addition but probably not necessary.

---

> ### Author Rebuttal · Authors · 2023-08-10
>
> Dear Reviewer W4cB,
>
> We sincerely appreciate your thoughtful review and your kind words about the novelty and potential significance of our work.
>
> "Extensions":
> We report models with simple patch embedder, we use cross-attention for image and audio, but not for text. We used strided inference only when comparing with prior published work (Tables 2 and 3). We will add all those details in our revised version. This would indeed make it easier for readers to follow, thanks!
>
> More details:
> We tried out bets to add
> For the local model, when not using cross-patch attention, we do use position embeddings; The GPT-3 architecture mentioned in the paper does not include the use of locally banded sparse attention patterns. We will add these details to the revised version.
>
> Reproducibility and code release:
> We plan to release the code associated with the paper and will make that clear in the revised paper.
>
> We appreciate your constructive feedback and agree with your suggestions for improvement. We will incorporate these changes in our revised paper to improve its clarity, detail, and reproducibility.
>
> Thank you for your positive evaluation and for considering our paper for acceptance.

---

> > ### Comment · Reviewer_W4cB · 2023-08-15
> >
> > Happy to hear the code will be released.

---

### Official Review · Reviewer_RJHZ · 2023-07-06

**Soundness:** 3 good
**Presentation:** 3 good
**Contribution:** 3 good
**Rating:** 7
**Confidence:** 5

**Summary:**

A new auto-regressive decoder-only Transformer variant for language modeling is presented which operates well on byte level. Similar to the Vision Transformer, the main idea is to operate on patches instead of single tokens (bytes). Those patches are composed of a fixed number of $P$ tokens. Then, a big global Transformer operates on those patches, i.e. on the spatial dimension $T / P$. After that, a smaller local Transformer operates on the spatial dimension $P$ inside a patch.

Actual values for $P$ are quite small, e.g. 8 for language modeling.

The model is tested on text for classic language modeling, but also on image data and audio data, where PPL on byte-level is measured in all cases. The performance on language modeling compared to other BPE-level models is worse. When comparing to other models on byte-level, it is always better or at least as good.

**Strengths:**

- Novel model which performs well on byte-level
- Many experiments, including ablations.


**Weaknesses:**

- The argumentation on sub-quadratic computation runtime for self-attention is flawed and not true. See details below.
- Proposed model can also be used on BPE-level. How would this perform? Maybe even with this model, BPE would be prefered? This experiment is missing.

**Questions:**

Really sub-quadratic? How? From the figure 1 and 2 it just looks like reduced by constant factor P or thus P^2?

"MegaByte decomposes long sequences into two shorter sequences, and optimal patch sizes reduces the self-attention cost to $O(N^{\frac{4}{3}})$" - I don't understand this. Why two shorter sequences? Where do I see that in the model? What is meant by optimal patch size? $P$ is the patch size? As I understand it, $P$ is a fixed hyper parameter, which you cannot change later on, as it is the dimension for the input.

"By setting $P = T^{\frac{1}{3}}$ - this does not make sense. $T$ is a variable length, depending on the input. $P$ is a fixed hyperparameter, which you can set once and then not change anymore. Or if you want to say, you also never change $T$ in practice, then the whole cost reduces to $O(1)$, but it's not really meaningful to think of $T$ as a fixed size.



Line 136: "the Global model has a sequence of length $\frac{P}{T}$ - I assume it should be $\frac{T}{P}$.
Line 136: "Local model uses $\frac{P}{T}$ sequences" - I assume it should be $\frac{T}{P}$ as well?

"Extensive experiments show that MEGABYTE allows byte-level models to perform competitively with subword models on long context language modeling" - but not on standard language modeling tasks?


$E^{\textrm{global-pad}} \in \mathbb{R}^{P \times D_G}$ should probably be $E^{\textrm{global-pad}} \in \mathbb{R}^{P \cdot D_G}$ instead?

"Section ?? proposes strided inference as a solution" - cross reference broken.

From the NeurIPS template: "The table number and title always appear before the table". However, this is wrong in the paper, where the table caption is incorrectly below the table.

Figure 4, Figure 7: I think that are tables, not figures?

Table 1 caption should describe, what are those numbers for the models? PPL?

In Table 1, Stories dataset, for PerceiverAR, the number looks inconsistent to all other numbers in that it only has 2 digits after the decimal point instead of 3. Is this on purpose?

Image modeling: It is not stated, but the model operates on the raw bytes, and you have 4 bytes per pixel? This should be explained.

The proposed model can be used to also operate on BPE level instead of byte level. While this is against the motivation of the work, I think this is still an interesting experiment. And also an important experiment. Because I would expect that this still improves over the byte-level variant. And this implies that BPE tokens are still overall better, and it is probably better to use BPEs also for this model, just like it is better to use BPEs for the other models.

**Limitations:**

-

---

> ### Author Rebuttal · Authors · 2023-08-10
>
> Dear Reviewer RJHZ,
>
> Thank you for your thoughtful review of our submission and for recognizing the novel aspects of our work. We appreciate your feedback and are eager to clarify the points you have raised:
>
> Sub-quadratic computation runtime argument:
> Thankyou for raising this issue, which is a bit subtle, and needs to be better described in the camera ready. There are really two places where T can be chosen for a Transformer model –- before training (which determines the maximum training sequence length, the number of positional encodings, and MegaByte's patch size), and during inference. Our analysis is about the former, whereas you assume the latter. We argue the former is more meaningful, because it is well established that during inference, most transformers generalize badly to sequences longer than they were trained on --- meaning that the training sequence length is usually an upper bound on the inference sequence length, so we cannot simply apply a pre-trained model to arbitrarily long sequences. However, the only limits on scaling to a large T during training are computational, so asymptotic analysis is more relevant here.
>
> In our experiments, we do scale the patch sizes for problems where we need to model longer sequences (although we do not use the theoretical optimal choice). This scalability is what allows us to model longer sequences than are possible with vanilla transformers.
>
>
> Use of model on BPE-level:
> Your suggestion to experiment with our model on the BPE-level is indeed an interesting one, and we agree it could provide additional insights. Our byte-level experiments are grounded on our initial observations that most byte predictions are relatively easy (hence, can be handled by a small local model) due to the locality within a patch. We think it needs some extensive studies to apply it to BPE tokens and beyond the scope of this work. However, we are extremely interested in how our model works on BPE-level and such an experiment could potentially further demonstrate the versatility of our proposed model. We will study this in our future work.
>
> Various formatting issues:
> We'll fix the reference and table format in the revised version. Thanks for your feedback, this makes the paper  cleaner to readers. For image modeling, the model does operate on the raw bytes from three different channels, so we have 3 bytes for each pixel. We will add a clear description in the revised paper.
>
> We greatly appreciate your constructive feedback, which will significantly improve the quality of our paper. We look forward to incorporating your suggestions in our revised version.

---

### Official Review · Reviewer_bsD9 · 2023-07-06

**Soundness:** 3 good
**Presentation:** 2 fair
**Contribution:** 3 good
**Rating:** 6
**Confidence:** 4

**Summary:**

This paper proposes Megabyte, a Transformer architecture that combines global and local attention to scale sub-quadratically in sequence length. The authors evaluate Megabyte on byte-level language modeling, image modeling, and audio modeling.

**Strengths:**

There is an extensive empirical validation, and Megabyte seems to perform well in the evaluated contexts. Megabyte has strong performance across different modalities, and scales well to long sequence lengths. The analysis of FLOP cost is nuanced and refreshing to see in an ML paper. Overall strong work.

**Weaknesses:**

There are a few ways that the paper could be improved.

First, the presentation of the paper is a bit messy. There are links/references that go across page boundaries (e.g., bottom of page 5 in the submission). There are also some dead references (e.g. line 304). These problems should be cleaned up in the camera ready.

The analysis of whether the language models can use their full context seems a little naive. A better analysis would be to see if perplexity on long documents (e.g., books) is worse than perplexity on short documents, relative to a baseline.

It would be nice to see some comparison against state-of-the-art tokenizer-based language models, e.g. GPT-Neo [1], Pythia [2], Hyena [3], H3 [4]. These models are not byte-based, but they are standard benchmarks of Transformer quality for language modeling.

Lastly, it would be nice to see if this approach scales to larger models.

[1] https://github.com/EleutherAI/gpt-neo
[2] https://github.com/EleutherAI/pythia
[3] https://arxiv.org/abs/2302.10866
[4] https://arxiv.org/abs/2212.14052

**Questions:**

1. Can you conduct a perplexity-based analysis of whether Megabyte uses the full context of its models?
2. Can you compare against state-of-the-art tokenizer-based language models?
3. Does Megabyte scale?

**Limitations:**

Brief mentions of scaling limitations but no concrete experiments

---

> ### Author Rebuttal · Authors · 2023-08-10
>
> Dear Reviewer bsD9,
> We appreciate the time you have taken to review our paper and provide us with constructive feedback. We agree with your suggestions to improve the paper, and will address your concerns below:
>
> Presentation and references:
> We apologize for the presentation issues you've mentioned, such as the cross-page references and dead links. We will rectify these issues in the final version of the paper to ensure a cleaner, more reader-friendly format.
>
> Megabyte uses the full context of its models:
> Figure 5 gives a perplexity-based analysis if MegaByte's ability to use its full context window on PG19. The average likelihood of a token increases monotonically throughout the 8192 token context window, which we argue shows that the model is able to use 8k previous tokens to improve its predictions.
>
> Comparison with state-of-the-art tokenizer-based language models:
> We focus on controlled comparisons between architectures, and it is difficult to make meaningful scientific comparisons with language models trained with very different sizes, datasets and compute budgets. For example, we might outperform some open source models on books data simply because of the kind of data we trained on. We do however show competitive results with SOTA tokenizer-based models trained on the PG19 corpus (better on dev, worse on test).
>
> Scalability of Megabyte:
> Some experiments in the paper are already at quite large scale, for example the SOTA runs on image64 takes about a week on 128 A100s. These results suggest MeagByte scales well, but training at true foundation model scale is beyond the scope of this submission, and we are happy to study that in the future work.
>
> We appreciate your constructive feedback and agree with your suggestions for improvement. We will incorporate these changes to improve the overall quality and impact of our paper.
> Thank you for considering our paper for acceptance.

---

> > ### Comment · Reviewer_bsD9 · 2023-08-16
> >
> > Thank you for answering my questions.

---

### Official Review · Reviewer_tnBu · 2023-07-07

**Soundness:** 4 excellent
**Presentation:** 4 excellent
**Contribution:** 4 excellent
**Rating:** 8
**Confidence:** 4

**Summary:**

This paper proposes a new architecture of sequential modeling called MEGABYTE. MEGABYTE splits the long data sequence into patches and encode the patches using the Global model. The output of Global model is forwarded into the Local model, which facilitates the within-patch encoding. The output of the Local model is used to make next position generation. Compared with typical sequence modeling architectures, MEGABYTE has the advantages of no tokenization, lower complexity of attention, much larger feedforward layers for the same computation and faster generation speed via higher parallelisms. The experimental results over various benchmarks shows the effectiveness of this model design. Sufficient analysis and ablations are also provided.

**Strengths:**

1. This paper focuses on a crucial problem in current sequential modeling tasks, which is making the encoding of extreme long sequences more efficient.
2. The architecture is easy to understand and reasonable.
3. The experiments are conducted on prevalent benchmarks related to long sequence modeling. And the SOTA models are compared on these benchmarks.
4. Several tricks are proposed to improve the performance of MEGABYTE, including strided inference, cross-patch attention and convolutional embedding. The contribution of these tricks are also analyzed in detail.
5. The models are implemented using open-sourced framework, which make it more reproducible.

**Weaknesses:**

Generally speaking, I think this paper is complete. I would say if more case study and analysis can be provided (especially on the language modeling task), it will be very good for a byte-level sequential model.

A typo: Line 220: datasest → dataset

**Questions:**

Just an extra question, is there any experimental results with the use of FLASH-attention and other SOTA speed and memory optimization libraries, given the fact that they are prevalently used in sequential models? Will the experimental conclusions still holds the same?

**Limitations:**

This work proposed a fundamental model architecture for efficient byte-level sequence modeling. This research topic is crucial for the development of future large language models. From my perspective, I would say there is not any so-called negative societal impact in this work.

---

> ### Author Rebuttal · Authors · 2023-08-10
>
> Dear Reviewer tnBu,
>
> We greatly appreciate your positive and thoughtful review of our paper. We're glad you acknowledged our contributions and the effort we made in ensuring reproducibility. We would like to address your comments and questions:
>
> Typo Correction:
> We appreciate you pointing out the typo. It will be corrected in the final version of the paper.
>
> More Case Studies and Analysis:
> We'd be happy to add further case studies, and would welcome any suggestions here.
>
> Experimental Results with FLASH-Attention and Other Libraries:
> Your question about the application of FLASH-attention and other state-of-the-art speed and memory optimization libraries is an important one. Flash-attention, particularly, is an exact attention algorithm that uses tiling to reduce the number of memory reads/writes between GPU high bandwidth memory (HBM) and GPU on-chip SRAM. Adding flash-attention, it should (theoretically) improve megabyte and also maintain its advantage compared with naive transformers. However, many SOTA speed and memory optimization libraries (including Flash-attention) are not implemented or tested for long sequences or more attention heads, requiring writing new cuda-kernels. We will add  FLASH-attention and other state-of-the-art speed and memory optimization libraries in the future work.
>
> In conclusion, we thank you for your encouraging review and your constructive suggestions. We believe your feedback will help improve the clarity and impact of our paper.

---

> > ### Comment · Reviewer_tnBu · 2023-08-20
> >
> > Thank the authors for providing the response. I will keep the rating as strong accept.

---

### Decision · Program_Chairs · 2023-09-21

**Decision:**

Accept (poster)

**Comment:**

The paper proposes an improved transformer architecture that by having local transformer to decoder byte level token more efficiently. The reviewers are generally in favor of the acceptance, acknowledging the novelty in the paper, and good experimental results on byte-level modeling of text, images and audio. Particularly, the work shows a good promise for language modeling without needing tokenization. However, there are some limitations and doubts of the proposed method. Such as the byte-level performance is still lag behind compared to BPE-based models and it is unclear if the proposed method helps when BPE is used. There are also some concerns around details and clarity in the paper, along with reproducibility concerns. Overall the reviewers assessed the submission positively and recommended acceptance after the authors addressed the feedback.